# The Effects of Etchant on via Hole Taper Angle and Selectivity in Selective Laser Etching

**DOI:** 10.3390/mi15030320

**Published:** 2024-02-25

**Authors:** Jonghyeok Kim, Byungjoo Kim, Jiyeon Choi, Sanghoon Ahn

**Affiliations:** 1Department of Laser & Electron Beam Technologies, Korea Institute of Machinery & Materials, 156 Gajeongbuk-Ro, Yuseong-Gu, Daejeon 34103, Republic of Korea; imj02096@kimm.re.kr (J.K.); byungjookim@kimm.re.kr (B.K.); jchoi@kimm.re.kr (J.C.); 2Department of Mechanical Engineering (Robot∙Manufacturing Systems), University of Science and Technology, 217 Gajeong-Ro, Yuseong-Gu, Daejeon 34113, Republic of Korea

**Keywords:** selective laser etching, chemical etching, selectivity, taper angle, ultrashort pulsed laser

## Abstract

This research focuses on the manufacturing of a glass interposer that has gone through glass via (TGV) connection holes. Glass has unique properties that make it suitable for 3D integrated circuit (IC) interposers, which include low permittivity, high transparency, and adjustable thermal expansion coefficient. To date, various studies have suggested numerous techniques to generate holes in glass. In this study, we adopt the selective laser etching (SLE) technique. SLE consists of two processes: local modification via an ultrashort pulsed laser and chemical etching. In our previous study, we found that the process speed can be enhanced by changing the local modification method. For further enhancement in the process speed, in this study, we focus on the chemical etching process. In particular, we try to find a proper etchant for TGV formation. Here, four different etchants (HF, KOH, NaOH, and NH_4_F) are compared in order to improve the etching speed. For a quantitative comparison, we adopt the concept of selectivity. The results show that NH_4_F has the highest selectivity; therefore, we can tentatively claim that it is a promising candidate etchant for generating TGV. In addition, we also observe a taper angle variation according to the etchant used. The results show that the taper angle of the hole is dependent on the concentration of the etchant as well as the etchant itself. These results may be applicable to various industrial fields that aim to adjust the taper angle of holes.

## 1. Introduction

Recently, in order to enhance IC chip performance, flip chip bonding has been widely adopted. To achieve flip chip bonding, numerous via holes are required. Therefore, a through-silicon via (TSV) has been applied. However, silicon has several disadvantages, such as its relatively high price and electric noise at high radio frequency. On the other hand, glass has unique properties that are suitable for interposer material, namely, low permittivity, high transparency, and adjustable thermal expansion coefficient. Signal noise can be avoided because of its low permittivity, three-dimensional alignment can be easily achieved because of its transparency, and warpage can be prevented because its thermal expansion can be matched with a Si wafer. Therefore, through-glass via (TGV) is becoming an increasingly popular alternative to TSV [1,2,3,4,5,6,7].

There are various methods for generating holes in glass [8]: mechanical drilling [9,10], powder blasting, abrasive slurry jet machining (ASJM) [11,12], laser drilling, deep reactive ion etching (DRIE) [13,14], plasma etching, spark-assisted chemical engraving (SACE), vibration-assisted micromachining, laser-induced plasma micromachining (LIPMM) [15], water-assisted micromachining, and selective laser etching [16]. The most important parameter to be considered in mass production is uniformity, although processing time should also be considered. Selective laser etching (SLE) can create uniform holes but currently requires a relatively long processing time.

Selective laser etching (SLE) is a widely used method for creating precise and intricate patterns on various materials. It was first introduced in 2001 by A. Marcinkevičius [17]. This technology involves the use of a laser and etchant to selectively remove material from the substrate, resulting in intricate patterns and shapes [17]. It can also be used to create holes in various materials, including metals, ceramics, silica, and glass [16,17,18]. It irradiates a laser to generate local modifications on the sample and proceeds with the etching process [17]. The modification area has about a 333 times higher etch rate compared to the non-modification area [16]. This is because the modification area changes in both physical and chemical properties, and the modification area quickly reacts with etchants. The physical and chemical changes include nanograting formation, volume expansion, and refractive index change [19,20]. These enable channel generation inside glass with excellent accuracy [17,21]. SLE is currently used in various fields such as biotechnology, nanotechnology, optics, and IT technology [16,18,21]. Our previous study shows how to enhance the process speed by changing the local modification method. We suggest that adding additional pulse energy after a few hundred picoseconds after the initial pulse can increase the etch rate of the local modification area [22]. In this study, we try to increase the etch rate with the etchant itself. To achieve this, four different etchants are tested, namely, HF solution, NaOH solution, KOH solution, and NH_4_F solution. For a quantitative comparison, the selectivity has been adopted. Our results show that NH_4_F solution has the highest selectivity and TGV can be formed within 3 h of etching with it. This is three times faster than a previous study that used KOH solution [22].

The glass holes generated via SLE can be also applied to various fields because the taper angle can be adjusted. The taper angle of glass holes is a crucial factor in several industrial applications. For instance, in the semiconductor industry, shower heads are used in the cleaning process, and the taper angle of the shower head holes determines the velocity and direction of the cleaning solution [23]. Similarly, in the field of biotechnology, microneedles with tapered holes are used to deliver drugs or extract fluids from the body. The taper angle of the holes affects the flow rate and penetration depth of the microneedle [24]. The taper angle of the hole also affects the spray pattern. This can be utilized in abrasive processes as well [25,26]. Therefore, research on the taper angle is useful for various industrial applications. In this study, we find that the taper angle varies according to the etchant itself and its concentration. Based on our study, we can tentatively claim that it is possible to increase productivity and adjust a hole’s taper angle based on the choice of etchant.

## 2. Experiments

### 2.1. Substrate Material

A borosilicate glass (D 263^®^ T eco, SCHOTT, Mainz, Germany) substrate with a thickness of 0.1 mm was used for the TGV process. This glass is a material that uses environmentally friendly cleaning agents instead of arsenic and antimony substances. The company that makes it claims that it has excellent chemical resistance and thermal stability and that it can be used in a wide range of applications that require high-precision parts. In particular, because of its thermal expansion characteristics, it is the best possible candidate material for glass interposers. Therefore, it is a suitable material for use in this study.

### 2.2. Ultrashort Pulsed Laser

Ultrashort pulsed lasers refers to lasers with pulse durations of one hundred picoseconds or less. An ultrashort pulsed laser has several advantages in glass processing. First, it transfers energy to the glass before thermal diffusion occurs, minimizing residue as a specific heat process and allowing microfabrication without post-processing steps. Second, it has high peak powers due to its short pulse durations, which makes it useful for processing materials with large energy bandgaps, such as glass. The high peak power of the ultrashort pulsed laser induces nonlinear absorption induction in the glass, enabling the internal processing of the glass.

### 2.3. Local Modification by Ultrashort Pulsed Laser

In this study, Bessel beams were used to increase productivity. Bessel beams have the advantage of a long depth of focus. Therefore, a single pulse of a Bessel beam can produce a full-thickness local modification in glass. Figure 1 is a schematic diagram of the experimental setup for local modification by an ultrashort pulsed laser. A diode-pumped Yb: KGW ultrashort pulse laser (Pharos, PH1-20, photoconversion, center wavelength 1030 nm) is used as the energy source. Our previous study confirmed that double pulses with a 213 ps interval are a suitable local modification process condition for generating TGV [22]. In addition, based on our previous study, a pulse duration of 1 ps is also adopted. The pulse repetition rate is 100 Hz and is synchronized with the motion stage (M-414.2PD, C-863.11, Physik Instrument, Karlsruhe, Germany). The stage movement speed is 10 mm/s. Therefore, the distance between each local modification is 100 μm. A pulse energy of at least 30 µJ is required to produce a local change. The pulse energy on the glass surface is measured and calculated via a power meter (NovaII, Ophir with 30A-BB-18 sensor, Ophir, Jerusalem, Israel). The total pulse energy reach to the sample is 68 μJ (34 μJ + 34 μJ).

### 2.4. Bessel Beam Shaping

Beam shaping from a Gaussian beam to a Bessel beam goes through two stages. First, the phase image is applied to the LCOS-SLM modulator (X10468-03, Hamamatsu Photonics, Shizuoka, Japan) to change the beam shape from a Gaussian beam to a donut beam. In this study, phase images are generated with optical engineering programs (VirtualLab Fusion, LightTrans, Jena, Germany). In our experimental setup, an image with 64 phase levels is sufficient to generate a donut beam. Next, the donut-shaped beam is focused with a plano-convex lens (25.4 mm focal length). Eventually, a Bessel beam is formed and measured with a beam profiler (FM100-YAG1064-50x, Metrolux, Berlin, Germany), where the Bessel beam width is 5.8 μm (FWHM) and the beam length is 180 μm.

### 2.5. Selective Laser Etching

In this study, we use the SLE technique to generate TGV. Normally, KOH solution is used to etch glass [27]. However, the purpose of this study is to enhance the SLE process speed by modifying the etching process. Thus, various solutions were tested, namely, HF solution, NaOH solution, KOH solution, and NH_4_F solution.

According to the Arrhenius equation, elevating the temperature of the etchant increases the frequency of collisions and accelerates chemical reactions. Therefore, the etching temperature was set to 10 °C below the boiling point of each etchant. For safety reasons, there was a 10 °C buffer.

After the local modification process with an ultrashort laser, the modified glass sample was immersed in a Teflon jar, which was filled with the etchant. Then, the jar was placed in an oil bath (WHB-6, Dai Han Scientific, Seoul, Republic of Korea) which was filled with a heat transfer fluid (Therminol D12^®^, Kingsport, TN, USA) for a certain amount of hours. As mentioned above, the etching process was performed at a suitable temperature for each etchant. After the etching process was finished, we cleaned the glass with DI water and IPA. Then, we placed the glass sample on the polyester wiper and removed the deionized (DI) water and isopropyl alcohol (IPA), as well as the remaining residue, with compressed nitrogen gas. This etching system uses a safety valve that keeps the internal pressure constant in the Teflon jar, as previously patented [28] (Figure 2).

## 3. Results and Discussion

### 3.1. Etchants and Selectivity

The goal of this study was to enhance TGV productivity. In this study, the SLE process was adopted. We generated a local modification inside the glass via an ultrashort pulsed laser and applied chemical etching. As mentioned above, four different etchants (HF, NaOH, KOH, and NH_4_F) were tested in order to increase the etching speed. 

For a quantitative comparison between each etchant, the concept of selectivity was adopted. Selectivity was calculated based on the etch rate of the modification area and non-modification area. Since each etchant has different characteristics, we decided to compare the fastest etching conditions. The calculated selectivity results are presented in Table 1 and Figure 3.
$$Selectivity=\frac{Modification\;area\;etch\;rate\;+\;Non\;Modification\;area\;etch\;rate}{Non\;Modification\;area\;etch\;rate}$$

For HF, etching was carried out at room temperature for 15 min with a 10 M solution. The etch rate of the modification area was measured as 191.0 µm/h, and the etch rate of the non-modification area was 143.9 µm/h. As a result, the selectivity of the HF 10 M solution was calculated as 2.3. Figure 4a shows the etching result for HF. Since the etch rate of HF was too fast and the selectivity was relatively low, a via hole could not be formed. 

For NaOH, etching was carried out at 90 °C for 20 h with a 3 M solution. The etch rate of the modification area was measured as 5.0 µm/h, and the etch rate of the non-modification area was 1.7 µm/h. As a result, the selectivity of the NaOH 3 M solution was calculated as 3.9. Figure 4b shows the etching result for NaOH. After 20 h of etching, hourglass-shaped via holes were generated.

For KOH, etching was carried out at 110 °C for 9 h with an 8 M solution. The etch rate of the modification area was measured as 12.5 µm/h, and the etch rate of the non-modification area was 4.1 µm/h. As a result, the selectivity of the KOH 8 M solution was calculated as 4.1. Figure 4c shows the etching result for KOH. After 9 h of etching, similar to NaOH cases, hourglass-shaped via holes were generated. However, the taper angle was different. We discuss this issue in Section 3.2.

For NH_4_F, etching was carried out at 85 °C for 3 h with an 8 M solution. The etch rate of the modification area was measured as 33.3 µm/h, and the etch rate of the non-modification area was 10.5 µm/h. As a result, the selectivity of the NH_4_F 8 M solution was calculated as 4.2. Figure 4d shows the etching result for NH_4_F. In the case of NH_4_F etching, it only takes 3 h to generate a via hole. Even though the etch rate of NH_4_F is the second fastest of the four etchants, it has the highest selectivity value. Thus, we can conclude that selectivity is a reasonable parameter for determining etching efficiency. We can also tentatively claim that the NH_4_F 8 M solution is the most efficient etchant for generating TGV of the four etchants. 

### 3.2. Etchants and Hole Taper Angle

As shown in Figure 4, taper angles are different for each etchant. Thus, we tried to establish the feasibility of changing the taper angle with the etchant and its concentration. Based on this study, we conclude that the taper angle can be determined by the molar concentration of each etchant (HF, NaOH, KOH, and NH_4_F). 

For the HF solution, 1 M (3.5 mL (49% HF solution) + 96.5 mL (deionized (DI) water)), 3 M (10.6 mL (49% HF solution) + 89.4 mL (deionized (DI) water)), 5 M (17.75 mL (49% HF solution) + 82.25 mL (deionized (DI) water)), and 10 M (35.5 mL (49% HF solution) + 64.5 mL (deionized (DI) water)) solutions were tested. Since the HF solution is a strong acid, the etching process was performed at room temperature for safety reasons. Moreover, the etch rate of the HF solution was too fast; therefore, the glass was dipped for only 10 min in order to observe the taper angle. The measured angles were 50° for 1 M solution, 53° for the 3 M solution, 53° for the 5 M solution (Figure 5a), and 41° for the 10 M solution. In addition, when the via hole was formed, a taper angle of 34° was measured. In this case, samples were etched in a 10 M HF solution for 15 min at room temperature. It was the lowest taper angle among all cases. The etching results are summarized in Figure 6.

In the case of the NH_4_F solution, 1 M (8.42 mL (40% NH_4_F solution) + 91.58 mL (deionized (DI) water)), 3 M (25.25 mL (40% NH_4_F solution) + 74.75 mL (deionized (DI) water)), 5 M (42.09 mL (40% NH_4_F solution) + 57.91 mL (deionized (DI) water)) (Figure 5b), 6 M (50.5 mL (40% NH_4_F solution) + 49.5 mL (deionized (DI) water)), 7 M (58.9 mL (40% NH_4_F solution + 41.1 mL (deionized (DI) water)), and 8 M (67.3 mL (40% NH_4_F solution) + 32.7 mL (deionized (DI) water)) solutions were tested. For a 40% NH_4_F solution, the boiling point is 109 °C, so for safety reasons, etching was performed at 85 °C for 2 h. The measured angles were 47° for the 1 M solution, 55° for the 3 M solution, 53° for the 5 M solution, 53° for the 6 M solution, 53° for the 7 M solution, and 50° for the 8 M solution (Figure 6). When a via hole was formed, a taper angle of 60° was measured. In this case, samples were etched in an 8 M solution for 3 h at 85 °C. We expected a higher etch rate to have a higher taper angle. However, it is independent. The most efficient etching solution had the second lowest taper angle (47°~60°).

In the case of the NaOH solution, 1 M (13 mL (20% NaOH solution) + 87 mL (deionized (DI) water)), 2 M (26 mL (20% NaOH solution) + 74 mL (deionized (DI) water)), 3 M (40 mL (20% NaOH solution) + 60 mL (deionized (DI) water)), 4 M (53 mL (20% NaOH solution) + 47 mL (deionized (DI) water)), and 5 M (100 mL (20% NaOH solution) solutions were tested. A 20% NaOH solution has a boiling point of 110 °C; thus, for safety reasons, etching was performed at 100 °C for 3 h. The measured angles were 58° for the 1 M solution, 57° for the 2 M solution, 62° for the 3 M solution (Figure 5c), 55° for the 4 M solution, and 53° for the 5 M solution. When a via hole was formed, a taper angle of 55° was measured (Figure 6). In this case, samples were etched in a 3 M solution for 20 h. 

In the case of the KOH solution, 1 M (8.5 mL (45% KOH solution) + 91.5 mL (deionized (DI) water)), 2 M (17 mL (45%KOH solution) + 83 mL (deionized (DI) water)), 3 M (25 mL (45% KOH solution) + 75 mL (deionized (DI) water)), 4 M (34 mL (45% KOH solution) + 66 mL (deionized (DI) water)), 5 M (42 mL (45% KOH solution) + 58 mL (deionized (DI) water)), 8 M (68 mL (45% KOH solution) + 32 mL (deionized (DI) water)), and 10 M (85 mL (45% KOH solution) + 15 mL (deionized (DI) water)) were tested. For a 45% KOH solution, the boiling point is 132 °C, so for safety reasons, the 1–5 M solution was etched at 100 °C, the 8 M solution at 110 °C, and the 10 M solution at 115 °C for 5 h. The measured angles were 58° for the 1 M solution, 66° for the 2 M solution, 60° for the 3 M solution, 63° for the 4 M solution, 64° for the 5 M solution, 63° for the 8 M solution, and 58° for the 10 M solution (Figure 5d and Figure 6). When a via hole was formed, a taper angle of 63° was measured. In this case, samples were etched in an 8 M solution for 9 h at 110 °C. The results indicated that KOH has the highest taper angle. 

In Figure 6, it can be observed that HF has the lowest taper angle and KOH has the highest taper angle. In our experimental setup, a sample was etched by HF at room temperature, NH_4_F at 85 °C, NaOH at 100 °C, and KOH at 100–115 °C. Thus, temperature might also have an effect on determining the taper angle other than etchant. Therefore, we performed etching by using the KOH solution with different temperatures. At 100 °C, a taper angle of 58–64° was measured, while a 58° taper angle was measured at 115 °C. Thus, we can conclude that temperature does not determine the taper angle. In addition, the results indicate that the concentration of the etchant determines the taper angle. However, there is no trend or linear relationship between the concentration and the taper angle (Figure 7). As shown in Figure 7, there is a certain concentration that generates the highest taper angle, and it varies with each etchant. However, in most cases, the highest taper angle is generated at the 3–5 M concentration of each etchant, except for KOH. 

## 4. Conclusions

This study presents the etching results according to four different etchants and etching conditions. The four etchants are HF, NH_4_F, NaOH, and KOH solutions, and the etching conditions are the concentration of each etchant and the etching temperature. We conducted this study to find the optimal process condition to enhance the productivity of TGV in glass interposers. The results can be summarized as follows:(1)The results show that the most efficient etchant is NH_4_F, and the TGV could be generated within 3 h through etching with the 8 M NH_4_F solution at 85 °C. As mentioned above, we could generate TGV three times faster than demonstrated in previous studies. In addition, we found that selectivity is the most trustworthy parameter for representing etching efficiency.(2)The results also reveal that the taper angle of a blind hole is affected by the etchant. The etchant itself determines the taper angle. HF, NH_4_F, NaOH, and KOH solutions generated 41°–53°, 47°–60°, 53°–62°, and 58°–66° taper angles, respectively. This study might be helpful for those who want to generate holes with certain angles. However, we still need to understand the principle underlying the phenomena identified in this study.

## Figures and Tables

**Figure 1 micromachines-15-00320-f001:**
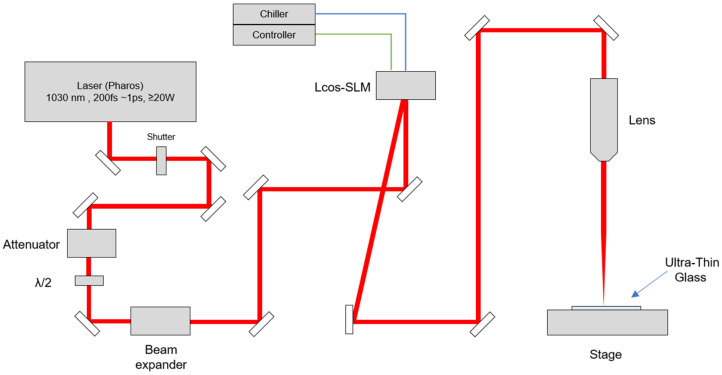
Schematic diagram of the experimental setup for local modification via an ultrashort pulsed laser.

**Figure 2 micromachines-15-00320-f002:**
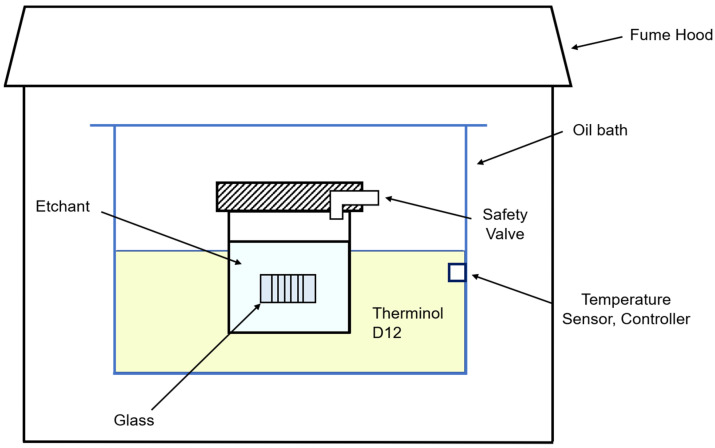
Schematic diagram of the etching process [28].

**Figure 3 micromachines-15-00320-f003:**
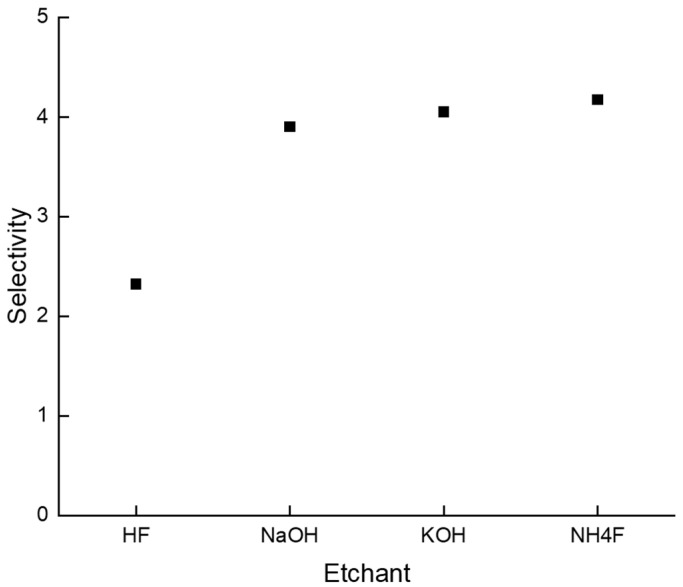
Selectivity of HF, NaOH, KOH, and NH_4_F solutions.

**Figure 4 micromachines-15-00320-f004:**
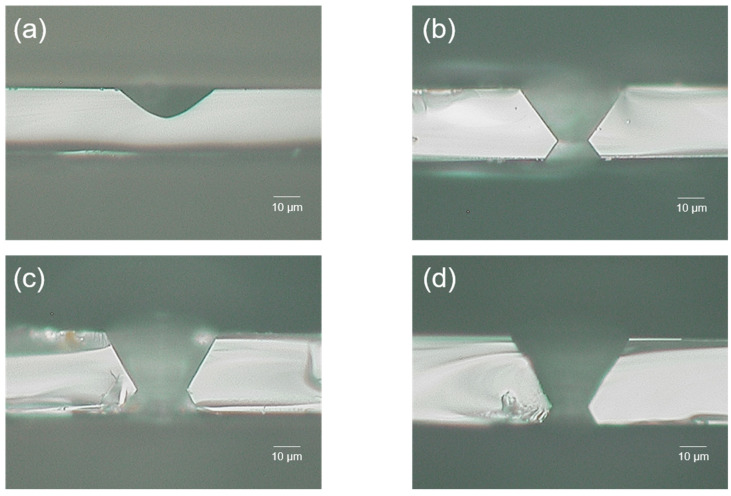
Via hole formed by each etchant. (**a**–**d**) Optical microscope images. (**a**) HF 10 M solution; (**b**) NaOH 3 M solution; (**c**) KOH 8 M solution; (**d**) NH_4_F 8 M solution.

**Figure 5 micromachines-15-00320-f005:**
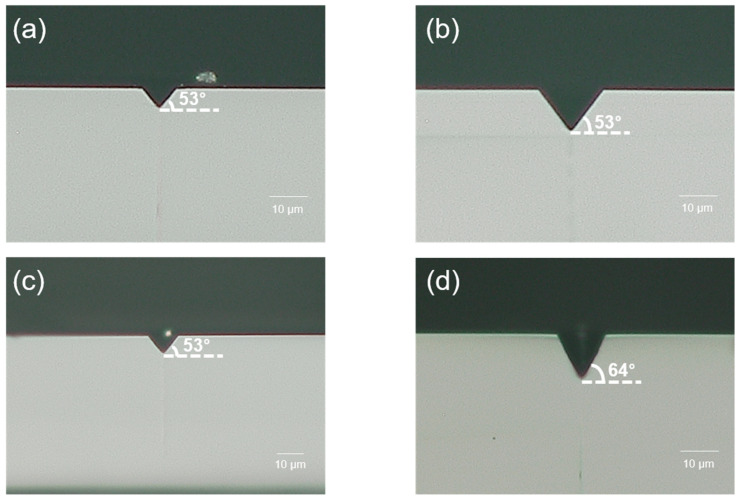
Via hole formed via each etchant. (**a**–**d**) Optical microscope images. (**a**) HF 5 M solution; (**b**) NH_4_F 5 M solution; (**c**) NaOH 5 M solution; (**d**) KOH 5 M solution.

**Figure 6 micromachines-15-00320-f006:**
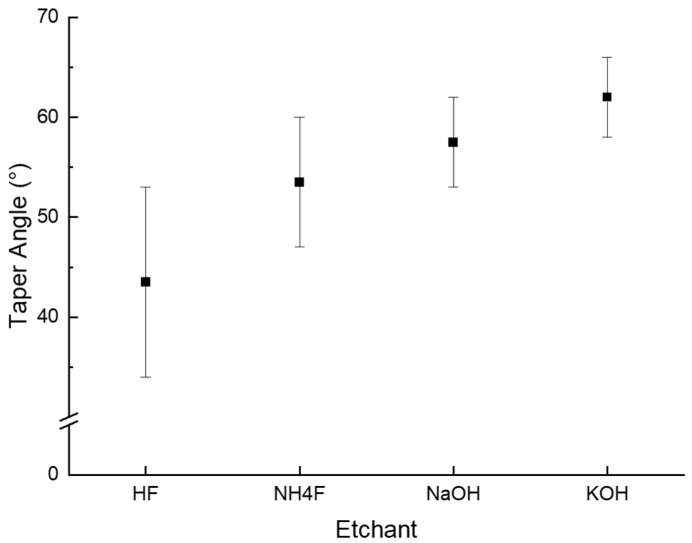
Taper angle range of HF, NaOH, KOH, and NH_4_F solutions.

**Figure 7 micromachines-15-00320-f007:**
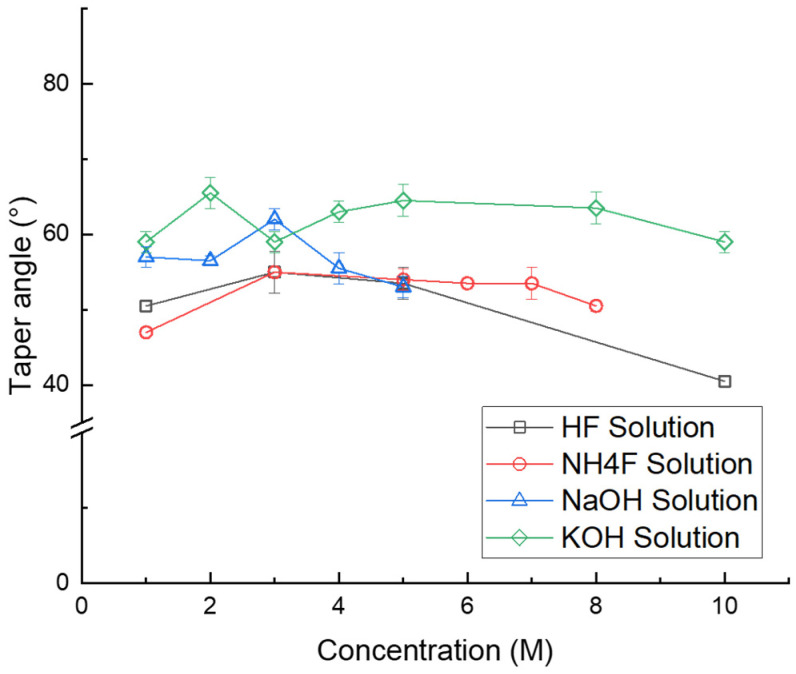
Taper angle based on concentration of HF, NaOH, KOH, and NH_4_F solutions.

**Table 1 micromachines-15-00320-t001:** Etching conditions and etch rate and selectivity for each etchant.

Etchant	Modification Etch Rate (μm/h)	Non-Modification Etch Rate (μm/h)	Selectivity
HF	191.0	143.9	2.3
NaOH	5.0	1.7	3.9
KOH	12.5	4.1	4.1
NH_4_F	33.3	10.5	4.2

## Data Availability

The data underlying the results presented in this paper are not publicly available at this time but may be obtained from the authors upon reasonable request.

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
