# Peer review of "The Effects of Etchant on via Hole Taper Angle and Selectivity in Selective Laser Etching"

_micromachines, 2024, doi:10.3390/mi15030320_

Round 1
Reviewer 1 Report
Comments and Suggestions for Authors
The title of the manuscript is "The Effects of Etchant on Via Hole Taper Angle and Selectivity in Selective Laser Etching", but it is limited to individual experiments and lacks systematic experiments. This article also lacks a dynamic analysis and model on the relationship between corrosion and pore formation.
Author Response
Thank you for valuable comments. The main purpose of this study is the comparison between various etchant which can be applied to mass production line. We also agree with your comment that the study does not include any numerical model or chemical analysis. Because this topic (improvement of process speed in TGV formation process) is hot issue in semiconductor industry, we try to submit this study as soon as possible. Later on, we will study more deeply.
Reviewer 2 Report
Comments and Suggestions for Authors
1.Please explain the specific application of borosilicate glass in the field of chips.
2.The test results are perfect. Please explain all the results fully.
3.The font size in the diagram needs to be adjusted.
4. Please divide the conclusion into sections.
Comments on the Quality of English LanguageModerate editing of English language required
Author Response
Thank you very much for your kind comments. We changed our paper as your recommendations.
1.Please explain the specific application of borosilicate glass in the field of chips.
-> We changed the use of borosilicate glass at line 86-87. Because of its thermal expansion characteristics, it is the most possible candidate for interposer material in semiconductor industry.
2.The test results are perfect. Please explain all the results fully.
-> We already explain about our results through section 3. Can you specify which result should be explained more?
3.The font size in the diagram needs to be adjusted.
-> A font size of figure 2 has been changed. (line 144-145)
- Please divide the conclusion into sections.
-> We divide the conclusion to two sections. (line 268-283)
Reviewer 3 Report
Comments and Suggestions for Authors
In this manuscript, the authors conducted experimental investigations on the effects of etchant on the via-hole formation and evaluated the selectivity of the etchant in the selective laser etching process. The results have been obtained with carefully designed experiments. However, in-depth discussion is missed, and some conclusions are not solid based on the experimental results. To make this manuscript qualified for publication, the authors are required to address the following comments well.
1. In line 11 and 12, please replace TGV and IC with their full names.
2. The quality of English writing could be better. Throughout the manuscript, there are many grammatical errors in the full text of the paper where subordinate clauses are incorrectly presented as main clauses. The authors are strongly recommended to rewrite the manuscript to correct grammatical errors and increase the variety of sentence structures and expressions.
3. In line 41, “chipping” should be “engraving”.
4. The reference citation format should be modified. They are supposed to be included before the period rather than after it.
5. Please add SPACE between value and unit throughout the manuscript.
6. The results shown in Figure 4 are obtained in different concentrations from the descriptions on page 5. Please clarify the reason. In addition, on page 5, etchants were assigned to Figure 4, with a different sequence shown in Figure 4.
7. Please mark the taper angle in the figures.
8. Please use “Figure #” instead of “figure #”.
9. The font size in line 273 and 274 is inconsistent with the rest of the manuscript. Please modify them.
10. Sentences from line 274 to line 276 need to be modified to correct the typo and grammatical issues.
11. What is the new knowledge that this manuscript brings about?
12. Please add references on line 67.
13. If provided with a longer etching time, will the taper angle become larger or not?
14. Based on Figure 7, it’s challenging to say that the taper angle is affected by concentration. Those angles fluctuate around one certain value. In addition, how many samples were prepared for the taper angle measurement? Each measurement should be repeated to get a solid conclusion, and then the mean value and standard deviation value should be provided.
15. A couple of hours is too much for drilling a hole with a depth of 0.1 mm on glass. Laser ablation/drilling can make it faster. Please address the significance of chemical etching, which is more desired here.
16. The content of Figure 3 is duplicating part of the information in Table 1. Please remove it.
Based on the abovementioned comments, this manuscript is recommended for major revision. A revised manuscript is required.
Comments on the Quality of English Language
Please check the comments for the authors.
Author Response
Thank you very much for your kind and detailed comments. It is really helpful to enhance the quality of our paper. Here are our answers of your valuable comments.
- In line 11 and 12, please replace TGV and IC with their full names.
-> It has been changed. “through glass via (TGV)” and “integrated circuit (IC)”
- The quality of English writing could be better. Throughout the manuscript, there are many grammatical errors in the full text of the paper where subordinate clauses are incorrectly presented as main clauses. The authors are strongly recommended to rewrite the manuscript to correct grammatical errors and increase the variety of sentence structures and expressions.
->the title of section 2 has been changed. “2. Experiments”
- In line 41, “chipping” should be “engraving”.
-> It has been changed. “spark-assisted chemical engraving (SACE)”
- The reference citation format should be modified. They are supposed to be included before the period rather than after it.
-> It has been changed. (Ref #1, 2, 4, 5, 6, 10, 18, 23, 28)
- Please add SPACE between value and unit throughout the manuscript.
-> It has been changed. (Line 104, 107-109, 112, 122, 124, 125, 166-168, 171-173, 176-178, 182-184, 189, 192,193, 201-203, 207-209, 212-216, 219-221, 224-227, 229-230, 232-242, 248, 249, 262)
- The results shown in Figure 4 are obtained in different concentrations from the descriptions on page
-> It has been changed. “(a) HF 10 M solution, (b) NaOH 3 M solution, (c) KOH 8 M solution, (d) NH4F 8 M solution”
- Please clarify the reason. In addition, on page 5, etchants were assigned to Figure 4, with a different sequence shown in Figure 4.
-> We changed the sequence at Figure 4 (HF, NaOH, KOH, NH4F)
- Please mark the taper angle in the figures.
-> It has been added on figure 5.
- Please use “Figure #” instead of “figure #”.
-> It has been changed. (line 101, 114, 145, 158, 163, 168, 174, 179, 185, 192, 197, 208, 211, 214, 220, 229, 230, 242, 248, 251, 252, 260, 265)
- The font size in line 273 and 274 is inconsistent with the rest of the manuscript. Please modify them.
-> It has been changed through the manuscript
- Sentences from line 274 to line 276 need to be modified to correct the typo and grammatical issues.
-> The conclusion section has been rewritten.
- What is the new knowledge that this manuscript brings about?
-> Before this paper, NH4F has not be applied on SLE process. In addition, we found that the etchant itself decide the taper angle.
- Please add references on line 67.
-> It has been added (Ref. #22). “. It is 3 times faster than previous study which used KOH solution. [22]”
- If provided with a longer etching time, will the taper angle become larger or not?
-> Not
- Based on Figure 7, it’s challenging to say that the taper angle is affected by concentration. Those angles fluctuate around one certain value. In addition, how many samples were prepared for the taper angle measurement? Each measurement should be repeated to get a solid conclusion, and then the mean value and standard deviation value should be provided.
-> We omit a deviation in the figure7 because it is too complicate to see. The taper angle test has been performed for 3 months and we haven’t count number of samples. It was abundant for conclude that etchant itself decides a taper angle.
- A couple of hours is too much for drilling a hole with a depth of 0.1 mm on glass. Laser ablation/drilling can make it faster. Please address the significance of chemical etching, which is more desired here.
-> We generated 20,000 local modification per second and etching it for 3 hours. If you use direct ablation to generate 20,000 holes, it will take more than 3 hours.
- The content of Figure 3 is duplicating part of the information in Table 1. Please remove it.
-> No, the table 1 includes etching rate thus it differs with figure3.
Round 2
Reviewer 3 Report
Comments and Suggestions for Authors
Most of the comments have been addressed well. However, the quality of English language is still not good. Throughout the manuscript, there are many grammatical errors in the full text of the paper where subordinate clauses are incorrectly presented as main clauses. The authors are strongly recommended to rewrite the manuscript to correct grammatical errors and increase the variety of sentence structures and expressions.
Comments on the Quality of English LanguageCheck the comments for authors.
Author Response
Thank you very much for comments.
We used english editing service from MDPI and those changes are highlighted as blue.